# Multifunctionality of Jasmonic Acid Accumulation during Aphid Infestation in Altering the Plant Physiological Traits That Suppress the Plant Defenses in Wheat Cultivar XN979

**DOI:** 10.3390/insects14070622

**Published:** 2023-07-11

**Authors:** Xia Yan, Shicai Xu, Jiao Guo, Jiazhen Hu, Dejia He, Li Jia, Huanzhang Shang, Guangwei Li, Kun Luo

**Affiliations:** Shaanxi Key Laboratory of Chinese Jujube, College of Life Science, Yan’an University, Yan’an 716000, China

**Keywords:** aphid infestation, aphid preference, life table, phytohormone signaling, transcriptional profiles

## Abstract

**Simple Summary:**

Our previous study demonstrated that previous aphid infestation significantly upregulates the transcription of marker genes associated with salicylic acid (SA) and jasmonic acid (JA) biosynthesis. However, accumulating experimental evidence has revealed that SA and JA signaling cascades in plant immune responses always act antagonistically across a diverse range of herbivore–plant systems. In this study, we determined the potential role of JA in suppressing plant defense responses in seedlings of the wheat cultivar XN979 through preference and performance experiments using grain aphids and evaluated the underlying mechanisms through the transcriptional profiles of marker genes related to JA- and SA-dependent responses and the formation of tryptamine. The obtained results reveal that the accumulation of JA during aphid infestation probably facilitates the forming of tryptamine, which is then probably converted into IAA or serotonin to alter the plant’s physiological properties and facilitate the feeding behaviors of aphids, compensating for the adverse effects of SA- or JA-dependent responses in host plants to colonizer attacks. As a result of this experiment, we have a better understanding of the antagonistic interactions between hormone signaling processes.

**Abstract:**

Crop plants have coevolved phytohormone-mediated defenses to combat and/or repel their colonizers. The present study determined the effects of jasmonic acid (JA) accumulation during aphid infestation on the preference and performance of *Sitobion miscanthi* Takahashi (Hemiptera: Aphididae), and its potential role in fine-tuning hormone-dependent responses in XN979 wheat cultivar seedlings was evaluated via the transcriptional profiles of marker genes related to JA- and salicylic acid (SA)-dependent responses. The preference experiment and the life table data reveal that direct foliage spraying of 2.5 mM methyl jasmonate (MeJA) exhibited weak negative or positive effects on the preferential selection and the population dynamics and oviposition parameters of *S. miscanthi*. The transcription level of phytohormone biosynthesis genes shows that foliage spraying of MeJA significantly upregulated the marker genes in the JA biosynthesis pathway while downregulating the SA pathway. In addition, either MeJA treatment or previous aphid infestation significantly induced upregulated transcription of the genes involved in the JA- and SA-dependent defense responses, and the transcription level of the tryptophan decarboxylase (*TaTDC*) gene, which facilitates the conversion of L-tryptophan to tryptamine, was rapidly upregulated after the treatments as well. The main products of tryptamine conversion could play a crucial role in suppressing SA-dependent defense responses. These results will provide more experimental evidence to enable understanding of the antagonistic interaction between hormone signaling processes in cereals under aphid infestation.

## 1. Introduction

Common wheat (*Triticum aestivum* L.) is one of the most widely cultivated plants in the world; however, its production is often threatened by a variety of wheat pests that seriously affect grain yield and quality [1,2]. The wheat aphid, *Sitobion miscanthi* Takahashi (Hemiptera: Aphididae), is one of the most destructive and common piercing–sucking herbivorous insects that seriously attacks cereal plants by directly sucking up photoassimilates and transmitting crop viruses such as barley yellow dwarf virus in the field agroecosystem, which has long been reported as *Sitobion avenae* (Fab.) in China [3]. For alleviating the damage caused by pest colonization, crop plants have coevolved intricate and dynamic immune defenses to combat further damage and/or repel subsequent colonizers [4,5,6,7]. Among these defense responses, phytohormones and their mediated signaling cascades play a critical role in triggering these defense responses against various species of pests [8,9,10,11]. For instance, the direct and indirect defense responses mediated by the phytohormone salicylic acid (SA) are the most important plant defenses against piercing–sucking insects, whereas jasmonic acid (JA)-mediated defense responses are always associated with resistance to chewing insects [12,13,14,15,16]. However, accumulating experimental evidence demonstrates that cereal plants preinfested with cereal aphids elevate the expression of many genes involved in the biosynthesis of SA and JA as well as SA- and JA-dependent defenses against further attacks [15,17,18]. Over the past several decades, the phytohormone biosynthesis pathways and their mediated signal transduction pathways, as well as their underlying molecular mechanisms, have been well documented in many herbivore–plant systems [2]. In addition, SA and JA signaling cascades in plant immune responses mostly act antagonistically across a diverse range of herbivore–plant systems [2,19,20,21].

During the past few decades, a diversity of regulators were found to be involved in the antagonistic interaction between SA and JA signaling cascades [2,19]. However, the potential roles of JA in the suppression of the defense responses triggered by SA signaling have been understudied [2,18,22,23,24]. In addition, experimental evidence in wheat–fungi and rice–planthopper systems suggests that the extensive accumulation of JA in plant tissue after attacks could lead to the production of compounds including serotonin and/or auxins [25,26,27]. Our latest review carefully reviewed the possible biosynthesis pathways of these compounds and their potential functions in fine-tuning plant defenses in cereal plants [24]. For instance, serotonin in rice could positively affect the performance of herbivore insects [25], while auxins enhance the pathogenicity of phytopathogens through alterations of plant physiological traits [11,26,28]. Although these data point in an important direction for exactly characterizing the antagonistic interaction between JA and SA signaling pathways in wheat plants, the direct or indirect role of JA in fine-tuning plant defense responses after exogenous application of JA or infestation with cereal aphids is underdetermined.

Therefore, the present study evaluated the effects of direct foliage spraying of methyl jasmonate (MeJA) on the preference and performance of *S. miscanthi*; subsequently, the expression profiles of candidate genes contributing to phytohormone biosynthesis and defense responses were determined. These results provide more experimental evidence on the antagonistic interaction between hormone signaling processes and more molecular targets for breeding elite wheat cultivars to control the damage caused by *S. miscanthi* in a sustainable manner.

## 2. Materials and Methods

### 2.1. Test Insect Maintenance

The apterous and alatae *S. miscanthi* specimens used in the following experiments were descendants of aphids that were originally collected from a winter wheat field around Yangling (37°14′ N, 108°04′ E), Shaanxi Province, China. The aphids were maintained on seedlings of the wheat cultivar (cv.) XN979 in a growth chamber under the previously described conditions [18] at the laboratory of insect physiology and ecology of Yan’an University. Prior to conducting the infestation experiment, the aphid specimens were reared for a couple of generations of stable reproduction to avoid biotic and abiotic effects. Approximately 10 adult apterous *S. miscanthi* were individually transferred to the newly cultured wheat seedlings in their three-leaf stage every two weeks and watered regularly.

### 2.2. Plant Growth Conditions

The winter wheat cv. XN979 used throughout the study was purchased from a seed market store located in Yangling, Shaanxi Province, China. Prior to cultivation, the wheat seeds with similar size and moisture content were soaked in deionized–distilled (dd) H_2_O for 1 day to germination. Then, several germinated seeds were sown in each plastic pot (9 × 9 × 10 cm) filled with a soil and seedling substrate (purchased from Zhonghe Agricultural, N-P-K ≥ 2%, organic matter ≥ 40%, conductivity ≤ 3, pH = 6–7) mixture at a 1:1 ratio and grown in the greenhouse conditions described previously at the laboratory of insect physiology and ecology, Yan’an University, Shaanxi Province, China. After planting, each pot was individually covered with a transparent insect-rearing cage (height 30 cm, diameter 8 cm) with a fine mesh (200 nylon mesh sieve) to prevent biotic effects. The wheat seedlings were watered regularly as needed from the bottom, and no further fertilizer was used during the experiment. The wheat plant developmental stage was described using the BBCH-scale [29,30]. During the one-leaf stage (BBCH 7), only one seedling, each of a similar plant size, was kept in each pot, and the other extra seedlings were individually removed from each pot. Wheat seedlings in the three-leaf stage (BBCH 13) were used in the experiments.

### 2.3. The Preference of S. miscanthi for Phytohormone-Treated Hosts

When the wheat plants reached the three-leaf stage (BBCH 13), the winter wheat seedlings of cv. XN979 were either exogenously sprayed with 2.5 mM MeJA or 2.5 mM SA (both of them containing Triton X-100 at 0.05%) or exogenously sprayed with ddH_2_O containing Triton X-100 at 0.05%, with these plants serving as the controls. The phytohormone compounds used in the following experiments, MeJA (methyl 3-oxo-2-(pent-2-enyl)-cyclopentaneacetate, 95%) and SA (2-hydroxybenzoic acid, 99.5%), were purchased from Shanghai Macklin Biochemical Technology Co., Ltd., China. Each treatment was set to 10 biological replicates. After drying for 6 h, the wheat plants undergoing different phytohormone treatments were randomly transferred into four different insect-rearing cages (60 × 50 × 45 cm) covered with a fine mesh (200 nylon mesh sieve) to allow for ventilation, and each cage had 15 pots of seedlings and 5 pots per treatment. A day before the infestation, more than 500 nearly adult aphids from the monoclonal population of apterous or alatae *S. miscanthi* were collected and placed into the insect-rearing boxes (containing wheat leaf tissues). Aphids that reached the adult stage within 24 h were considered ready for the preference experiment. Prior to artificial infestation, the aphids were starved for 3 h. For each seedling, 10 apterous or alatae *S. miscanthi* within 24 h of reaching the adult stage were individually transferred to the leaves of wheat plants in different infestation experiments. The number of adult aphids and aphid nymphs produced by these adults colonizing the shoots of each plant were recorded at 12 h, 36 h, 60 h, 84 h, 108 h, and 132 h following aphid infestation, and the newborn nymphs were removed after recording. Ten biological replicates were conducted for each treatment.

### 2.4. The Performance of S. miscanthi on Seedlings Treated with Phytohormones

An age–stage, two-sex life table was employed to assess the effects of direct foliage application of phytohormones on the performance of *S. miscanthi*. The final concentrations of phytohormones and the treatment protocol are the same as previously described in the preference experiments, and the life table experiments were conducted in the controlled greenhouse conditions described previously. After the phytohormone treatments, newly born apterous *S. miscanthi* nymphs (within 24 h after birth) were individually transferred onto the first leaf of the wheat plant in each treatment using a fine-hair brush. Daily observations were made, and larval molt, larval mortality, and time of reaching the adult stage were recorded. From the onset of reproduction to death, the number of newborn nymphs was recorded daily, and then the nymphs were removed. The population dynamics parameters of *S. miscanthi*, including the net reproductive rate (*R*_0_), intrinsic rate of natural increase (*r*), finite rate of increase (*λ*), and mean generation time (*T*), were evaluated from the raw data of the life table to determine the effects of direct application of phytohormones on the performance of *S. miscanthi*. The definitions, explanations, and equations of these parameters used in the life table analysis were described in [31], and the parameters of *S. miscanthi* were calculated and analyzed using the TWOSEX-MSChart software (Ver. 04/26/2022) [32]. In addition, several oviposition parameters, namely, adult preoviposition period, total preoviposition period, oviposition days, oviposition periods, daily larvae production during oviposition period, and fecundity of *S. miscanthi*, were considered to determine the effects of phytohormone treatments on their reproduction. Each treatment in the life table analysis was set to 30 biological replicates.

### 2.5. Gene Expression Profiles in Wheat Plants after Phytohormone Treatments

To elucidate any possible antagonistic interactions between phytohormone signaling processes in wheat plants after aphid infestation, five apterous *S. miscanthi* adults (starved for 3 h) were individually transferred onto the wheat leaves, and these aphids were removed 24 h after the initial infestation. The phytohormone treatment protocol for the plants was as described previously, while seedlings without exposure to aphids or phytohormones served as control plants. For detection of the gene transcriptional profiles in wheat plants after phytohormone treatments, approximately 0.5 g of foliar tissue of the third leaf was harvested in biological triplicates at 3 h, 6 h, and 9 h after phytohormone treatments, while leaf tissue in aphid infestation treatments was collected at 0 h, 6 h, 12 h, and 24 h after aphid removal, and control plant tissues were collected in biological triplicate at the same time points. All collected leaf tissues were frozen in liquid nitrogen and stored in a −80 °C freezer for total RNA extraction. Total shoot RNA was isolated from approximately 0.5 g of wheat leaves collected from each treated plant by using a total RNA extraction kit and strictly following the instructions in the reference manual (Solarbio Biotech, Beijing, China). The total shoot RNA was dissolved in RNase-free water (Solarbio Biotech, Beijing, China) and stored in a −80 °C freezer until assessment of the gene expression profiles. After ascertaining the purity, quality, and integrity of the total RNA via a spectrophotometer (Thermo Scientific, Waltham, MA, USA) and agarose gel electrophoresis, approximately 1 μg total RNA in each sample was reverse-synthesized for first-strand cDNA via EasyScript^®^ One-Step gDNA Removal and cDNA Synthesis SuperMix (TransGene Biotech Co., Ltd., Beijing, China) using the instructions in the reference manual. After diluting the reverse transcription reaction mixture five times, it was time to determine the gene expression profiles. The marker genes associated with the JA or SA signaling pathways were considered in the present study. Among them, the lipoxygenase (LOX) and allene oxide synthase (AOS) genes are crucial for the biosynthesis of JA, and the WRKY3 and pathogenesis-related protein (PR)-4 and PDF1.2 genes usually play important roles in JA-mediated defense responses, while the phenylalanine ammonia lyase (PAL) gene is involved in SA synthesis, and the genes including PR-1 and nonexpressor of PR genes 1 (NPR1) are involved in SA-mediated defense responses. The *TaTDC* gene was used as the marker gene for determining the relationship between JA accumulation and serotonin or auxin biosynthesis because it is crucial for forming tryptamine from L-tryptophan (L-TRP). The *TaGAPDH* gene was used as an endogenous reference gene. The primers were synthesized by Sangon Biological Technology (Shanghai, China) and are listed in Appendix A. A kit (FastSYBR Mixture with High ROX reagent) purchased from Jiangsu Cowin Biotech Co., Ltd. (Taizhou, Jiangsu, China), was employed to assess the expression profiles of these genes. Real-time quantitative PCR (RT-qPCR) was performed using the StepOnePlus^TM^ real-time PCR system (ABI, Carlsbad, California, USA). The 20 μL reaction mixtures consisted of 3.0 μL first-strand cDNA reaction product (~30 ng), 10.0 μL 2 × FastSYBR Mixture (with High ROX Reference Dye), 0.4 μL of each primer pair (10 μM), and 6.2 μL of ddH_2_O. The expression profiles of each gene were expressed with its relative expression levels and calculated with the 2^−ΔΔCt^ method [33]. The relative expression level of the controls was arbitrarily set to 1. Three independent biological replicates were set for calculating the mean and standard error of the relative expression levels of each gene.

### 2.6. Statistical Analyses

The parameters of *S. miscanthi* in the preference experiment and the relative expression levels of marker genes in the wheat shoot tissues were calculated using Microsoft Excel (version 2010, Microsoft, Redmond, WA, USA). One-way analysis of variance (ANOVA) was used to compare the parameters involved in the preferences of *S. miscanthi* when they were reared on wheat plants after direct foliage spraying with different phytohormones at the same sampling timepoint. In experiments where a significant difference among the samples was found, multiple comparisons between treatments were performed using Student–Newman–Keuls (SNK) tests. The bootstrap technique with 100,000 resamplings was employed to estimate the standard errors (SEs) of the parameters involved in the performance experiment of *S. miscanthi* and compare the difference in the TWOSEX-MSChart software [31,32]. The independent-samples Student’s *t*-test was performed to compare relative gene expression levels. All the differences were considered statistically significant at *p <* 0.05. These statistical analyses were performed using the SPSS software (version 26.0, SPSS Inc., Chicago, IL, USA). All graphs were prepared using the GraphPad Prism 8.0 software (GraphPad Software, San Diego, CA, USA).

## 3. Results

### 3.1. Direct Foliage Treatment with Phytohormone Affected the Preference and Oviposition of S. miscanthi

Under controlled greenhouse conditions, after direct foliage spraying with either 2.5 mM MeJA or SA, the preference and oviposition of *S. miscanthi* were altered, with increases in sampling at the timepoints after phytohormone treatments and the apterous or alatae adult *S. miscanthi* exhibiting similar performance profiles (Figure 1). The greatest number of adult aphid colonizations was found in wheat seedlings treated with MeJA, while the lowest value was found in SA treatments. Based on a one-way ANOVA, differences in apterous aphid colonization were only observed at 132 h, where less colonization was observed on SA-sprayed plants than MeJA and control plants (Figure 1A, *F* = 4.599, *df* = 2, *p* = 0.019), while the mean number of adult aphid colonizations and their oviposition did not exhibit significant differences among phytohormone treatments at other sampling timepoints (Figure 1A,B). In addition, the oviposition of apterous adult aphids exhibited a similar profile to that of adult aphid colonization (Figure 1B), while the oviposition of alatae adult *S. miscanthi* exhibited a more complex tendency; additionally, the MeJA treatment had its largest values at the 36 h, 108 h, and 132 h sampling timepoints (Figure 1D).

### 3.2. MeJA Application Weakly Exerted Positive Effects on the Performance of S. miscanthi

Based on the data collected from the age–stage, two-sex life table of *S. miscanthi*, the population dynamics parameters of *S. miscanthi* were significantly adversely affected after SA application (*p* < 0.05); however, MeJA treatment did not exhibit significant differences when compared with controls (Figure 2). In particular, SA treatment resulted in significant reductions in the parameters *R*_0_, *r*, and *λ* of *S. miscanthi*, and the smallest values of these parameters were observed in SA treatments, with a significantly prolonged *T* parameter (Figure 2). Although the values of parameters *R*_0_, *r*, and *λ* of *S. miscanthi* reared on the plants with MeJA application were close to those of the aphids reared on controls, these parameters exhibited significant differences between different phytohormone treatments, and the largest values of the *R*_0_ parameter of *S. miscanthi* were found in the MeJA treatments. For the oviposition parameters of *S. miscanthi*, the direct foliage spraying of SA exhibited seriously adverse effects on these parameters as well (Table 1). In particular, different phytohormone treatments significantly increased the value of the adult preoviposition period and total preoviposition period of *S. miscanthi* when compared with the controls (*p* < 0.05), and the MeJA treatment exhibited smaller increases in these two parameters than SA treatment, while the parameters of oviposition days, oviposition period, and daily larvae production during the oviposition period of *S. miscanthi* did not exhibit significant differences between treatments (*p* > 0.05), and the parameters of oviposition days and oviposition period exhibited their largest values when *S. miscanthi* fed on the seedlings with MeJA treatment (Table 1). For the *F* parameter of *S. miscanthi*, although MeJA application did not exhibit a significant difference when compared with controls, it exhibited a significant difference between different phytohormone treatments, and the largest value of the *F* parameter of *S. miscanthi* was found to follow the MeJA treatments.

### 3.3. MeJA Treatments Significantly Elevated the Marker Genes Involved in JA Signaling in Plants

Direct foliage spraying of 2.5 mM MeJA on winter wheat cv. XN979 significantly upregulated the transcriptional levels of marker genes associated with JA biosynthesis and defense responses (Figure 3). In particular, the relative expression levels of the *LOX* and *AOS* genes involved in the JA biosynthesis pathway were significantly elevated at the three sampling timepoints, and both of them reached their highest values at 6 h postspraying and were increased 3.47-fold (*t*_6h_ = −10.492, *df* = 12.313, *p* < 0.01) and 4.10-fold (*t*_6h_ = −5.773, *df* = 8.756, *p* < 0.01) relative to the control plants, respectively. Thereafter, the transcriptional levels of these two genes were gradually decreased at 9 h postspraying (*t_LOX_* = −8.092, *df* = 12.688, *p* < 0.01; *t_AOS_* = −2.710, *df* = 10.576, *p* = 0.021). Similar to the increase in JA biosynthesis gene expression, the expression levels of the JA-dependent response genes, including the plant pathogenesis-related protein 4 (*PR-4*) and *WRKY3,* were significantly increased. However, the transcriptional profiles of these two genes and the JA biosynthesis genes were extremely different; for instance, the expression levels of *PR-4* (*t*_3h_ = −7.969, *df* = 8.012, *p* < 0.01) and *WRKY3* (*t*_3h_ = −7.455, *df* = 9.551, *p* < 0.01) exhibited their largest values at 3 h postspraying and gradually decreased at the following sampling timepoints. Based on our previous study, exogenous application of either MeJA or SA could significantly increase the expression level of *PR-1*; thus, in the present study, we evaluated the expression profiles of *PR-1* after direct MeJA application. While the expression level of the *PR-1* gene did not exhibit its largest value during the first sampling (*t*_3h_ = −6.742, *df* = 8.056, *p* < 0.01), at 6 h postspraying, its value reached the largest value, 86.39-fold that of the controls (*t*_6h_ = 8.727, *df* = 8, *p* < 0.01), and finally decreased to 23.80-fold (*t*_9h_ = −6.020, *df* = 8.016, *p* < 0.01) that of the controls at 9 h postspraying (Figure 4). These results suggest that an antagonistic interaction between the expression levels of *PR-1* and *PR-4* was detected.

### 3.4. MeJA Treatments Significantly Suppressed the Marker Genes Involved in SA Signaling, While Elevating Tryptamine Biosynthesis Gene

In contrast, direct foliage spraying of 2.5 mM MeJA on winter wheat cv. XN979 significantly suppressed the relative expression levels of the marker genes involved in the SA biosynthesis and defense responses (Figure 4). At 3 h postspraying, the transcription level of the *PAL* gene (*t*_3h_ = 11.273 *df* = 11.045, *p* < 0.01) associated with SA biosynthesis was seriously suppressed; thereafter, the suppression effects of MeJA application on SA biosynthesis declined gradually. Meanwhile, the transcription level of the *NPR1* gene (*t*_3h_ = 14.092, *df* = 8.120, *p* < 0.01; *t*_6h_ = 11.079, *df* = 8.138, *p* < 0.01; *t*_9h_ = 0.908, *df* = 9.613, *p* = 0.386) involved in the activation of SA-dependent responses was significantly inhibited as well, with similar transcription profiles. As described previously, the transcription level of the *PR-1* gene usually involved in SA-dependent responses was not immediately upregulated after MeJA application; the direct effects of MeJA treatment became weakened, its transcription level reaching a maximum at 6 h after MeJA spraying. In addition, the transcriptional pattern of the *TaTDC* gene that generally converts L-TRP to tryptamine revealed that MeJA application in wheat cv. XN979 significantly upregulated the expression of this gene (*t*_3h_ = −40.385, *df* = 16, *p* < 0.01; *t*_6h_ = −15.458, *df* = 10.745, *p* < 0.01; *t*_9h_ = −2.782, *df* = 8.185, *p* = 0.023), which would provide the substrate to form auxins or serotonin. Similarly to the JA-dependent response genes, the expression level of the *TaTDC* gene exhibited its largest value 3 h postspraying and gradually decreased at the following sampling timepoints (Figure 5A).

### 3.5. Aphid Infestation Significantly Triggered the Expression of SA- and JA-Dependent Response Genes

As expected, the marker genes associated with SA and JA biosynthesis were significantly upregulated in wheat cv. XN979 with preinfestion of *S. miscanthi* (Figure 6 and Figure 7). In comparison, the transcription level of JA biosynthesis genes, especially the *LOX* gene, was greater than that of the *PAL* gene that generally contributes to forming SA, and both of them returned to their normal expression levels at 24 h after aphid removal (Figure 6). Meanwhile, the expression levels of the *AOS* gene at all sampling timepoints (*t*_0h_ = −4.034, *df* = 16, *p* < 0.01; *t*_6h_ = −8.945, *df* = 16, *p* < 0.01; *t*_12h_ = −4.834, *df* = 16, *p* < 0.01; *t*_24h_ = −5.292, *df* = 10.264, *p* < 0.01) were significantly induced compared to the control, and it showed the largest upregulation at 6 h after aphid removal. In addition, although both SA- and JA-dependent response genes were significantly induced after aphid infestation for 24 h, their transcriptional patterns completely differed. Similarly, the expression level of the *PR-1* gene (*t*_0h_ = −6.093, *df* = 8.036, *p* < 0.01) exhibited its largest value at the first sampling timepoint after aphid removal, after which it rapidly decreased at the following sampling timepoints (Figure 7). Meanwhile, the transcriptional level of the marker genes associated with JA-dependent responses, including *WRKY3* (*t*_6h_ = −5.299, *df* = 8.906, *p* < 0.01), *PDF1.2* (*t*_6h_ = −12.702, *df* = 8.119, *p* < 0.01) and *PR−4* (*t*_6h_ = −5.163, *df* = 8.064, *p* < 0.01), showed their largest values at 6 h after aphid removal; subsequently, their expression levels gradually decreased (Figure 6). Moreover, similarly to direct MeJA treatments, the extensive accumulation of JA in tissues of wheat cv. XN979 plants triggered after infestation with *S. miscanthi* significantly upregulated the transcriptional profiles of the *TaTDC* gene as well (Figure 5B). Similarly, the relative expression level of the *TaTDC* gene was immediately upregulated after aphid removal (*t*_0h_ = −3.549, *df* = 8.024, *p* < 0.01), after which its relative expression level reached the largest value at 6 h after aphid removal (*t*_6h_ = −5.141, *df* = 8.02, *p* < 0.01). At the 12 h sampling timepoint, the transcription of the *TaTDC* gene rapidly decreased to 13.36-fold (*t*_12h_ = −6.011, *df* = 8.067, *p* < 0.01) that of the controls and was still downregulated at 24 h following *S. miscanthi* removal (*t*_24h_ = −3.116, *df* = 8.422, *p* = 0.130), when it reached its lowest value.

## 4. Discussion

Previous studies on diverse pest– or fungus–crop systems demonstrated that exogenous spraying of either MeJA or SA could be an alternative way to suppress the severity of plant disease epidemics and the population dynamics of pests [34,35]. In the present study, the results show that the preference and performance of *S. miscanthi* were adversely affected when the aphids were reared on wheat cv. XN979 plants with SA treatments; however, the MeJA treatments exerted weak negative or positive effects on selective preference and the parameters involved in the growth, development, and reproduction of aphids. Because our latest studies revealed that infestation with cereal aphids could induce the upregulation of marker genes involved in SA and JA biosynthesis and subsequently active dependent defenses in their hosts against further attacks [18], the accumulation of JA during aphid infestation therefore probably exerts multiple effects on plant physiological traits (Figure 8). For instance, the transcription of the *PAL* gene associated with SA biosynthesis was significantly downregulated in wheat seedlings with direct MeJA treatment, while the marker genes related to JA biosynthesis were significantly upregulated. In addition, the transcription of marker genes in both SA biosynthesis and JA biosynthesis pathways was significantly upregulated in the seedlings infested by *S. miscanthi*. These results suggest that the direct foliage spraying of MeJA on wheat cv. XN979 could alter the fitness of *S. miscanthi*, most likely by suppressing SA accumulation and its dependent responses.

Meanwhile, according to the transcriptional pattern of marker genes associated with JA-dependent responses, the transcription levels of the *WRKY3*, *PDF1.2*, and *PR-4* genes were significantly upregulated at the early sampling timepoints after direct MeJA treatment or *S. miscanthi* infestation. It is possible that these genes would trigger defense responses in plant tissues. Interestingly, the defense responses did not exert serious negative effects on preference and performance in *S. miscanthi* when they colonized seedlings with exogenous application of MeJA. Therefore, treatment with MeJA probably triggers alternative pathways that alter plant physiological properties to finetune the defense responses in wheat foliage tissues. For instance, the transcription of the *TaTDC* gene was severely upregulated at all sampling timepoints in host plants with either MeJA treatments or *S. miscanthi* infestation, and its relative expression level reached 13.45-fold higher than that of the controls at 3 h after MeJA spraying. Both studies on fungi and plants have revealed that *TDC* generally converts L-TRP to tryptamine in a carboxyl transformation reaction [26]. The accumulating experimental evidence demonstrates that infection by *F. graminearum* leads to an increased biosynthesis of L-TRP and derived compounds in wheat and other major cereal grain species [36,37,38]. Accordingly, we are tempted to speculate that either *F. graminearum* infection or aphid infestation would immediately induce the accumulation of JA in plant tissues, and the increase in JA level would trigger the production of L-TRP. However, our previous feeding experiment in vitro revealed that when supplying *F. graminearum* cultures with L-TRP, the indole tryptophol was immediately detected, and no auxin was detected during the entire feeding experiment [39]. Therefore, the accumulation of JA in plant tissue experiencing aphid infestation or infected with *F. graminearum* would trigger the upregulation of the *TaTDC* gene, and the expression products would significantly transform L-TRP to tryptamine [40].

Subsequently, tryptamine could be a possible substrate to form auxins or serotonin, which significantly alter plant physiological properties or directly benefit the feeding herbivorous insects [24,25,40]. For instance, our previous studies showed that fungi could utilize the tryptamine in the environment to form IAA [41]; this pathogen-elicited IAA probably induces the upregulation of the candidate genes associated with IAA biosynthesis in the host plants and causes significant accumulation of IAA [24]. The most important consequences of IAA accumulation in the tissue of host plants would be fast release of plasma nutrients and increased opportunities for successful penetration, which could facilitate colonization by attackers [42]. More experimental evidence from canola (*Brassica napus* L.) and its parasites demonstrated that infection with the clubroot pathogen, a biotrophic pathogen-induced SA pathway, significantly suppressed the growth of susceptible canola and impacted the fitness and oviposition of the bertha armyworm (*Mamestra configurata* Walker) in canola. It revealed that SA accumulating after pathogen infection probably suppresses JA accumulation and its positive role in plant growth [43]. In addition, IAA may attenuate SA-dependent responses, possibly by activating the expression of JA biosynthesis-related genes and JA-regulated defense genes, including *AOS*, *LOX2* (lipoxygenase2), and *VSP2* in plant seedlings [20,42]. This conclusion was supported by the experimental evidence of the transcriptional patterns of the *AOS* and *LOX* genes after direct MeJA treatment or aphid infestation. Based on this experimental evidence, these two phytohormones act synergistically to attenuate SA-dependent responses induced after pest colonization of host plants. More work is required to identify the exact relationship between the upregulation of the transcription of marker genes associated with JA biosynthesis and JA or IAA accumulation in host plants.

In addition, the experimental evidence acquired from wheat—*F. graminearum*, rice—*Nilaparvata lugens*, and rice—*Sogatella furcifera* systems reveals that tryptamine can be converted into serotonin by tryptamine 5-hydroxylase, which always involves one step [25,27]. Similarly, serotonin has antagonistic interactions with the SA-mediated plant defense response, probably because serotonin and SA are derived from the same precursor [25]. Therefore, the large amount of chorismate in wheat tissue would trigger the biosynthesis of serotonin, which would suppress the conversion of chorismate to SA and result in the attenuation of system-acquired responses in plant tissues. Moreover, serotonin is a critical neurotransmitter in mammals that also stimulates the behavior and immunity of insects; it is additionally involved in plant growth, development, and response to biotic and abiotic stresses [25]. The multifunctioning of serotonin in host tissues would positively affect the feeding behaviors of aphids and might trigger more aggressive attacks on wheat plants. This experimental evidence suggests that herbivorous insects and phytopathogens evolved complex mechanisms to fine-tune plant defenses and benefit themselves to establish their parasitic relationships with their hosts.

It was demonstrated that RNA interference (RNAi)-mediated knockdown of transcription of the target genes associated with the performance of cereal aphids in the crop host could be a potential approach for withstanding their attacks [2]. Currently, direct foliage spraying of dsRNA products or gently rubbing a recombinant virus carrying dsRNA onto the surface of wheat leaves are the crucial measures for temporarily strengthening the resistance of the crop host [2,44,45]. There have been considerable efforts focused on suppressing the transcription of the crucial genes involved in the performance and virulence of pests by adopting these two methods [45]. Thus, the transcriptional profiles of marker genes related to JA-dependent responses and the *TaTDC* gene in the present study will provide molecular targets for gene-silencing-based cultivar improvements after pest infestation. Silencing the transcription of any of these genes, such as *TaAOS*, *TaLOX*, and *TaTDC*, would significantly strengthen SA-dependent responses during aphid infestation and against subsequent infestations. Meanwhile, a reliable way to detect the potential roles of these genes in aphid–wheat interactions is required for developing sustainable aphid control measures. Therefore, in future experiments, we will employ the recombinant-virus-mediated gene silence (VIGS) approach to determine the potential biological roles of these interesting genes in positively influencing the performance of aphids.

The experimental evidence in the present study revealed that direct MeJA treatment significantly upregulated the transcription of the *PR-1* gene, which is usually associated with SA-dependent responses in host plants and could efficiently alleviate further damage caused by piercing–sucking insects or biotrophic pathogens. This is consistent with our previous study, where we found that the direct foliage spraying of either MeJA or SA immediately induced the upregulation of the *PR-1* gene in the winter wheat line 35-E4 with aphid resistance during the seedling stage, while the susceptible line 35-A20 did not experience upregulated transcription of the *PR-1* gene after MeJA treatment [18]. However, previous studies revealed that the winter wheat cv. XN979 is susceptible to grain aphids during its seedling stage [46]. Therefore, it is difficult to determine whether there is a relationship between *PR-1* upregulation and induced resistance and constitutive aphid resistance according to these experimental results. In the following study, we will determine the relation between *PR-1* upregulation and induced resistance and constitutive aphid resistance in diverse wheat cultivars.

In summary, the accumulation of JA during *S. miscanthi* infestation may facilitate the conversion of tryptamine to form IAA or serotonin to alter plant physiological properties or facilitate the feeding behaviors of aphids to compensate for the adverse effects of SA- or JA-dependent responses in host plants to colonizer attacks. The results of the current study will provide evidence to understand the induced molecular defense mechanisms of wheat plants against piercing–sucking insects.

## Figures and Tables

**Figure 1 insects-14-00622-f001:**
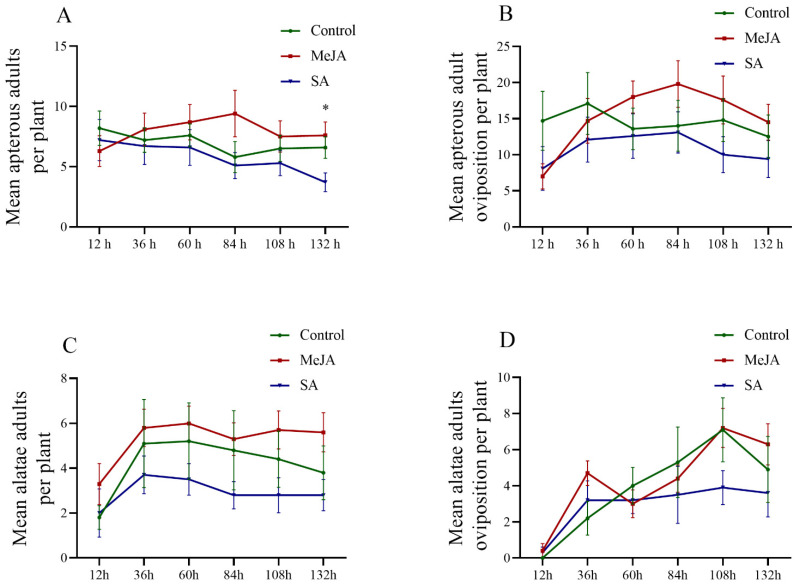
The colonization and oviposition (mean ± SE) of apterous or alate aphids on wheat cv. XN979 with direct foliage spraying with SA or MeJA phytohormones and control at 12 h, 36 h, 60 h, 84 h, 108 h, and 132 h after aphid infestation. (**A**) The mean number of apterous adult aphids; (**B**) the total number of ovipositions of apterous adult aphids; (**C**) the mean number of alatus adult aphids; (**D**) the total number of ovipositions of alatus aphids. * in Figure 1A indicate a significant difference in mean number of apterous adult aphids colonized on wheat plants between control and hormone spraying at 132 h timepoint by SNK test (*p* < 0.05).

**Figure 2 insects-14-00622-f002:**
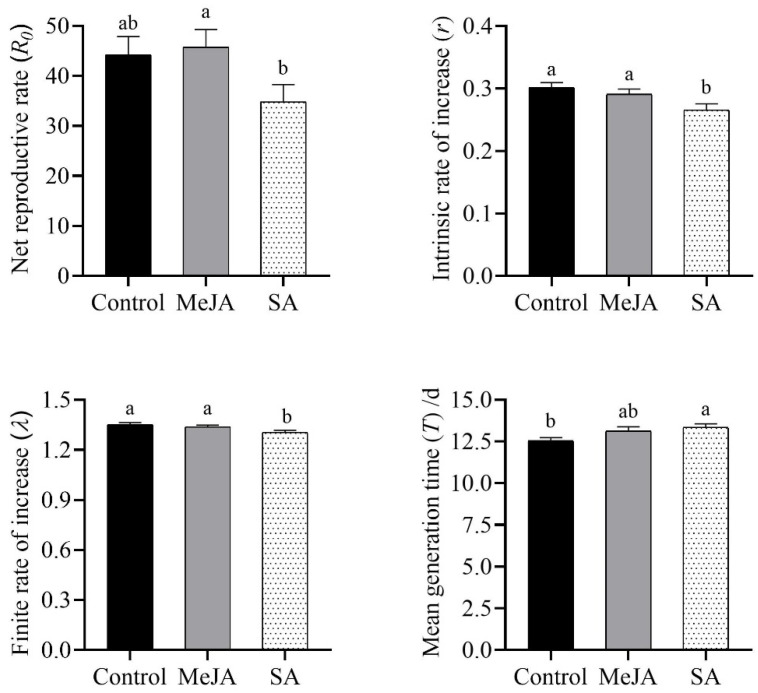
Effects of direct foliage treatment with different phytohormones on population dynamics parameters (mean ± SE) of *Sitobion miscanthi*. SEs were estimated by using the bootstrap technique with 100,000 resamplings. Different letters indicate significant differences (*p* < 0.05) between control and phytohormone spraying, determined by using the paired bootstrap test based on the confidence interval of difference at the 5% significance level.

**Figure 3 insects-14-00622-f003:**
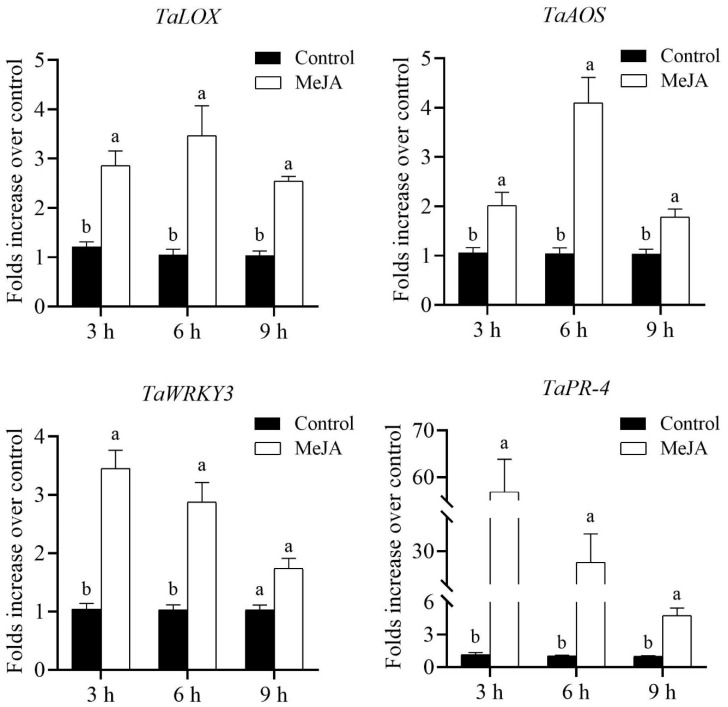
Expression profiles of marker genes associated with JA biosynthesis and its dependent responses caused by MeJA spraying on wheat cv. XN979 assayed via RT-qPCR during a time course of 9 h following JA treatment. The expression levels of plants without MeJA treatment were arbitrarily set to 1. The error bars correspond to the SEs. Different letters indicate significant differences in expression between control and hormone spraying at the same sampling timepoint by the independent-samples Student’s *t*-test (*p* < 0.05).

**Figure 4 insects-14-00622-f004:**
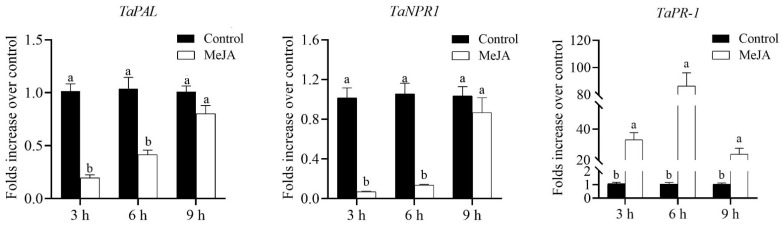
Transcriptional profiles of marker genes involved in SA biosynthesis and its dependent responses caused by hormone spraying in wheat cv. XN979 assayed via RT-qPCR during a time course of 9 h after MeJA treatment. The expression levels of plants without MeJA treatment were arbitrarily set to 1. The error bars correspond to the SEs. Different letters indicate significant differences in expression between control and hormone spraying in plants at the same sampling timepoint by the independent-samples Student’s *t*-test (*p* < 0.05).

**Figure 5 insects-14-00622-f005:**
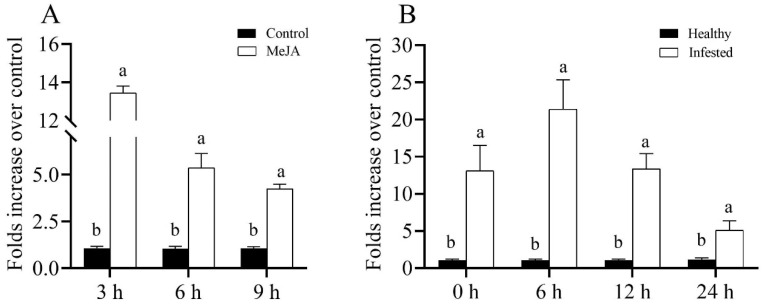
Expression profiles of genes associated with *TaTDC* induced by hormone spraying or *Sitobion miscanthi* infestation in wheat cv. XN979 seedlings assayed via RT-qPCR. The expression levels of plants without exposure to aphids or phytohormones were arbitrarily set to 1. The error bars correspond to the SEs. Different letters indicate significant differences in expression between control and preinfested plants at the same sampling timepoint by the independent-samples Student’s *t*-test (*p* < 0.05). The relative expression levels (**A**), after *S. miscanthi* infestation; (**B**), after 2.5 mM MeJA spraying.

**Figure 6 insects-14-00622-f006:**
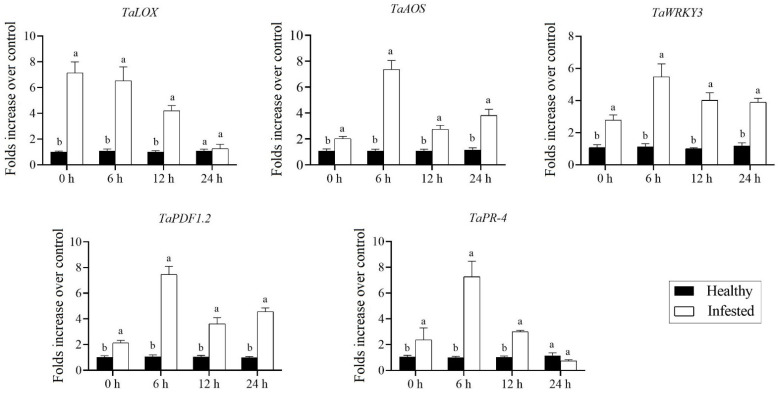
Expression profiles of marker genes associated with JA biosynthesis and dependent responses caused by *Sitobion miscanthi* infestation of wheat cv. XN979 assayed via RT-qPCR during a time course of 24 h after aphid removal. The expression levels of healthy plants were arbitrarily set to 1. The error bars correspond to the SEs. Different letters indicate significant differences in expression between control and preinfested plants at the same sampling timepoint by the independent-samples Student’s *t*-test (*p* < 0.05).

**Figure 7 insects-14-00622-f007:**
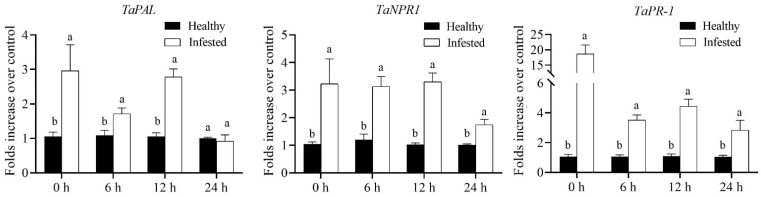
Transcriptional profiles of maker genes associated with SA-biosynthesis and SA-dependent responses caused by *Sitobion miscanthi* infestation in wheat cv. XN979 assayed via RT-qPCR during a time course of 24 h after aphid removal. The expression levels of healthy plants were arbitrarily set to 1. The error bars correspond to the SEs. Different letters indicate significant differences in expression between control and hormone spraying in plants at the same sampling timepoint by the independent-samples Student’s *t*-test (*p* < 0.05).

**Figure 8 insects-14-00622-f008:**
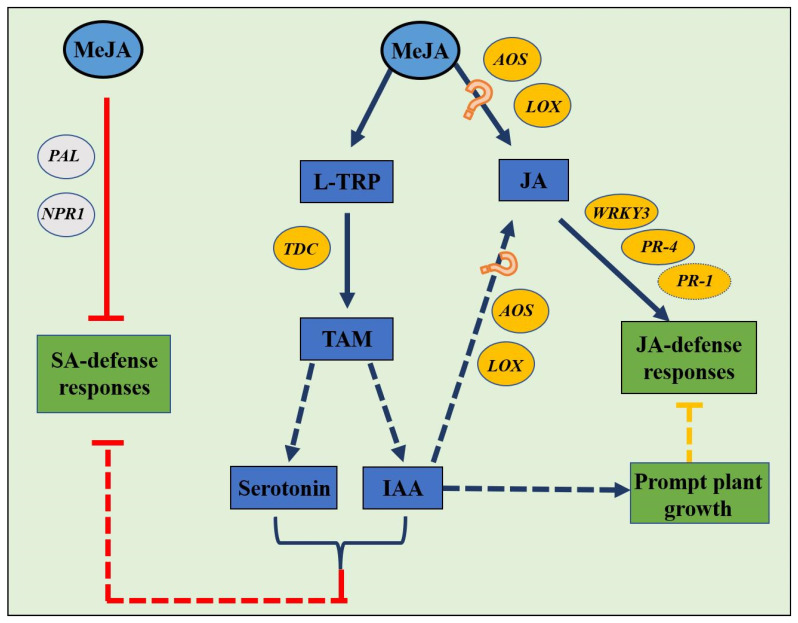
Proposed hormone-dependent response pathways induced by MeJA in wheat cv. XN979. Note: Filled arrows represent conversions observed in or deduced from this study, while the red blunt ends indicate a negative interaction (inhibition). The dotted arrows represent steps predicted from the literature, and the red question marks represent the pathways that we cannot exclude as possibilities from the results of the current study.

**Table 1 insects-14-00622-t001:** The oviposition parameters (mean ± SE) of *S. miscanthi* reared on wheat cultivar XN979 directly sprayed with different phytohormones.

Parameters	Treatments		
	Control	MeJA	SA
Adult preoviposition period, (d)	7.52 ± 0.11 ^c^	7.96 ± 0.14 ^b^	8.57 ± 0.24 ^a^
Total preoviposition period, (d)	0.60 ± 0.10 ^b^	0.89 ± 0.08 ^a^	1.25 ± 0.25 ^a^
Oviposition days, (d)	13.64 ± 0.90 ^a^	14.57 ± 0.63 ^a^	12.46 ± 0.77 ^a^
Oviposition period, (d)	14.28 ± 0.99 ^a^	15.57 ± 0.62 ^a^	13.18 ± 0.73 ^a^
Daily larvae production during oviposition period	3.30 ± 0.19 ^a^	3.13 ± 0.17 ^a^	2.75 ± 0.18 ^a^
Fecundity (*F*)	46.01 ± 3.32 ^ab^	49.03 ± 2.90 ^a^	37.30 ± 3.23 ^b^

Note: SEs were estimated by using the bootstrap technique with 100,000 resamplings. The means followed by different letters in the same column were significantly different (*p* < 0.05) according to the paired bootstrap test based on the confidence interval of differences at the 5% significance level.

## Data Availability

The datasets used or analyzed during the current study are available from the corresponding author upon reasonable request.

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
