# Peer review of "Multifunctionality of Jasmonic Acid Accumulation during Aphid Infestation in Altering the Plant Physiological Traits That Suppress the Plant Defenses in Wheat Cultivar XN979"

_insects, 2023, doi:10.3390/insects14070622_

Round 1

Reviewer 1 Report

"Previous Infestation with Sitobion miscanthi Induces Jasmonic Acid Accumulation that Suppresses Plant Defense Responses in Wheat Cultivar XN979."

In this article, Yan et al. first determined the aphid parameters (preference, performance, and reproduction) after foliage spraying of MeJA and/or SA on the wheat cultivar XN979 and further measured the induction of marker genes in the JA/SA pathways. Then, the authors demonstrated that aphid infestation, similarly to MeJA/SA spraying, induced the gene expression of the same marker genes in the pathways. While no clear conclusion was made from the results, the authors proposed that the accumulation of JA after aphid infestation could facilitate the conversion of tryptamine to IAA/Serotonin to mediate aphid feeding or SA and JA responses.

There are many uncertainties that could be explained and gaps that could be filled to improve the manuscript for further evaluation:

As the authors measured gene expression right after aphid infestation, I would rather use "infestation" and not "pre-infestation" throughout the manuscript.

Regarding the aphid preference on phytohormone-treated plants:

Line 115-131: The authors need to provide details on how to select the aphids within 24 hours of reaching the adult stage (lines 126-127). In addition, it is unclear how many experimental replicates were done for this experiment.

Figure 1: How can the possibility of the observed significance in panel A due to the variation in aphid ages be excluded? Denote the "*" clearly, as it only shows the significance between SA and the other treatments.

Regarding the aphid performance experiment:

Lines 133-150: The authors measured many aphid parameters, including molt, mortality, reproduction, F, r, T, APOP, TPP, etc., but without any explanations. The authors need to provide a brief explanation for all those parameters and ensure that it will be clear to readers how those parameters were measured. The same rule applies to Table 1. In addition, data for some of the mentioned parameters were not presented, such as molt, mortality, etc. It is also unclear how long the experiments were performed and especially at which time point the parameters in Figure 2 were measured. Furthermore, it is unclear how many experimental replicates were performed.

For the quantification of different gene markers:

For all the experiments and Figures 3-7: It is unclear why the authors only measured the gene expression up to 9 or 24 hours, which barely or does not overlap with the time point in the aphid performance experiments that were up to 132 hours. What is the gene expression status at those late hours, especially when significant differences are observed (Figure 1A)? Another question about those chosen time points is that at the endpoint (9 or 24 hours), most of the genes were still significantly upregulated in the treated samples compared to control. Why not measure the gene expression until they return to normal levels, as mentioned for the LOX and PAL genes (Lines 334-335)?

For Figures 6 and 7: Explain what "0h" represents. To my understanding, "0h" means without aphid treatment; however, all genes were significantly upregulated in the pre-infested group compared to the control. Please clarify.

In addition, it is unclear why the MeJA/SA treatments were separated from aphid infestation treatment in those gene expression measurements, and samples were collected at different time points. It would be better to have a side-by-side comparison of the gene expression between MeJA/SA treatments and aphid infestation.

For all experiments, overall discussion, and conclusion:

The authors have discussed how aphid infestation changes the gene expression in JA/SA signaling pathways and how those pathways could influence aphid performance, and potentially how the intermediates (auxins or serotonin) could be involved in the interactions between the host plant and aphids. However, none of those plant hormones were measured in any of the experiments. Measuring those hormones would bridge the gap between the current results and the authors' intended conclusions.

Figure 8 needs a higher resolution image, and the intermediates (chemicals such as serotonin, as proposed by the authors) need to be added to the model.

I hope my comments could be helpful in significantly improving the manuscript.

In general, the English could be moderately improved.

Author Response

Thank you very much for your letter and the comments about our manuscript submitted to your journal. We would also like to thank the reviewers for their careful and constructive reviews. Soon after receiving your comments, the other authors and I carefully revised the manuscript. The changes made according to the comments from the reviewers are detailed below.

#1

"Previous Infestation with Sitobion miscanthi Induces Jasmonic Acid Accumulation that Suppresses Plant Defense Responses in Wheat Cultivar XN979."

In this article, Yan et al. first determined the aphid parameters (preference, performance, and reproduction) after foliage spraying of MeJA and/or SA on the wheat cultivar XN979 and further measured the induction of marker genes in the JA/SA pathways. Then, the authors demonstrated that aphid infstation, similarly to MeJA/SA spraying, induced the gene expression of the same marker genes in the pathways. While no clear conclusion was made from the results, the authors proposed that the accumulation of JA after aphid infestation could facilitate the conversion of tryptamine to IAA/Serotonin to mediate aphid feeding or SA and JA responses.

Response 1: Thank you very much for carefully reviewing our manuscript and kindly give us many suggestions and comments for improving the readability of this manuscript. Yes, this study focused more on characterizing the performance and preference of grain aphids when exposure it to a certain concentration of phytohormones, such as salicylic acid (SA) and jasmonic acid (JA), which usually accumulated in plant tissues after aphids’ attack. The experimental evidence in many vegetables and tobacco demonstrated that either spraying or soaking a certain concentration of JA (around 2.5mM) could be severed as the effective measures to suppress the population of different species of pests. According to the results get in vegetable or tobacco, JA accumulated after aphid infestation should play the adverse effects on the performance and oviposition of subsequent aphid attackers, however, direct foliage spraying of 2.5 mM MeJA exhibited weak negative or positive effects on the preference and the parameters of S. miscanthi reared on wheat cultivar XN979, according to the preference experiment and the life-table data in this study. It looks that the results of this study have no specific outcome for aphid control in the agricultural production when adopt the phytohormone treatment. However, this study posed the important direction for characterizing the antagonistic interaction between hormone signaling in cereals. In addition, our previous study demonstrated that previous aphid infestation significantly upregulated the transcripts of genes associated with SA- and JA-biosynthesis, and subsequently it would trigger the hormones dependent responses. Although these defense responses play the crucial role in against the following attackers, a large amount of the aphids always stay together in the wheat leaves (Fig. 1), it looks that these defense responses didn’t exhibit the resistance to aphids damage. Thus, the experimental evidence get from this study would provide the important clue for understanding the reason and mechanisms why the aphids always stay together, and the heavy outbreak in each growing season in most of wheat production area. Moreover, the present study would provide the important molecular targets for cultivar improvement programs. In the discussion section, we pointed out the potential pathways for wheat cultivars breeding programs. In conclusions, we guess the findings of this study would provide the fundamental theory for developing the aphid management measures. About the one variety was used in the current study, the experimental evidence in wheat-fungi and rice-planthopper systems suggested that the extensive accumulation of JA in plant tissue after attacks could lead to the production of compounds, including serotonin and/or auxins in plant tissues, which would play the positive role in their damage. Accordingly, we guess the experimental evidence in the current study could explain the relationship between exogenous application of JA and the serotonin and/or auxins accumulation in common. Because this study focused more on understanding the biological roles of the physiological changes in wheat tissues after pests attacks to affected the fitness of subsequent colonizers. Thus, in the revision, we carefully modified the title of this manuscript based on the content of this study. Please allow us to modify the title of the manuscript.

Fig. 1 The different species of aphids stay together on the cereal leaves in different fields.

(Left: wheat leaves; Right: corn leaves)

There are many uncertainties that could be explained and gaps that could be filled to improve the manuscript for further evaluation:

Response: Yes, in the revision, we carefully followed your corrections.

As the authors measured gene expression right after aphid infestation, I would rather use "infestation" and not "pre-infestation" throughout the manuscript.

Response: Thank you very much for the rephrase suggestion. In the revision, we had replaced “pre-infestation” with “infestation” from main texts and the figures in the revision.

Regarding the aphid preference on phytohormone-treated plants:

Line 115-131: The authors need to provide details on how to select the aphids within 24 hours of reaching the adult stage (lines 126-127). In addition, it is unclear how many experimental replicates were done for this experiment.

Response: We apologize for not precisely described the selection methods of the apterous or alatae population of S. miscanthi with 24 h of reaching the adult stage. In the revision, we provided more details about how to conduct the aphid feeding experiment in lines 129-133. In addition, we didn’t point out the exact number of the biological replicates in the preference experiment in the original submission. When we conducted the preference experiment, each treatment was set the 10 biological replicates (ten pots per treatment), only one seedling with similar plant size was kept in each pot. After the wheat plants reaching the three-leaf stage, each seedling had individually transferred ten apterous or alatae S. miscanthi within 24 h of reaching the adult stage onto the leaves of wheat plants in different phytohormone treatment experiments. Then the number of adult aphid and aphid nymphs produced by these adults colonized on the shoots of each plant were recorded at different sampling timepoints post-aphid infestation. In the revision, we provided the above mentioned information in that section.

Figure 1: How can the possibility of the observed significance in panel A due to the variation in aphid ages be excluded? Denote the "*" clearly, as it only shows the significance between SA and the other treatments.

Response: Yes, it’s true that the variation of the apterous aphid colonization would be presented with the increasing of the aphid ages, however, according to the data collected from the present study, the wheat seedling with JA treatment had attracted the largest the number of adults aphid colonization at the most of sampling timepoints. Therefore, that tendency could reflect that JA treatment exhibited weak negative or positive effects on the preference of aphids. In addition, "*" in the Fig. 1 meant that on an one-way ANOVA, at 132 h post aphid infestation exhibited the significant difference between different treatments. Sorry for we didn’t compare the mean of aphid colonization per plant between different treatments.

Regarding the aphid performance experiment:

 Lines 133-150: The authors measured many aphid parameters, including molt, mortality, reproduction, F, r, T, APOP, TPP, etc., but without any explanations. The authors need to provide a brief explanation for all those parameters and ensure that it will be clear to readers how those parameters were measured. The same rule applies to Table 1. In addition, data for some of the mentioned parameters were not presented, such as molt, mortality, etc. It is also unclear how long the experiments were performed and especially at which time point the parameters in Figure 2 were measured. Furthermore, it is unclear how many experimental replicates were performed.

Response: We apologize for not precisely described the definitions, explanations, and equations of the parameters used in the life table analysis. Because there were a diverse of studies that referred those information, thus we didn’t illustrate them in the original submission. In the revision, as mentioned previously, for omitting the extra information, we cited a reference that illustrated these information in detail. Please allow us to make such modification in the revision. In addition, we recorded the larval molt, larval mortality, and the time of it reaching the adult stage and the number of nymphs of S. miscanthi, these raw data were used to calculate the parameters, including the net reproductive rate (R0), fecundity (F), intrinsic rate of natural increase (r), finite rate of increase (λ), and mean generation time (T). Therefore, these parameters could reflect the difference of molt and mortality of S. miscanthi between different treatments. Moreover, each treatment in the life table analysis were set the 30 biological replicates, and each aphids had the different life-span, thus the period of raw data collected and recorded is different and these parameters are the statistical value of the cohort population of 30 aphids. Finally, in the revision, we reconstructed the parameters in Fig. 2 and Table 1, please allow us to make such modification in the revision.

For the quantification of different gene markers:

For all the experiments and Figures 3-7: It is unclear why the authors only measured the gene expression up to 9 or 24 hours, which barely or does not overlap with the time point in the aphid performance experiments that were up to 132 hours. What is the gene expression status at those late hours, especially when significant differences are observed (Figure 1A)? Another question about those chosen time points is that at the endpoint (9 or 24 hours), most of the genes were still significantly upregulated in the treated samples compared to control. Why not measure the gene expression until they return to normal levels, as mentioned for the LOX and PAL genes (Lines 334-335)?

Response: As mentioned previously, we used several timepoints to illustrate the tendency of aphid colonization, which could reflect that JA treatment exhibited weak negative or positive effects on the preference of aphids. Because the limitation of movement speed of aphid, thus we extend the sampling timepoint to 132 h. Among of these sampling timepoints, most of them exhibited the similarly profiles on aphid colonization. Meanwhile, the relative expression of some marker genes had restored to normal levels with 24 h sampling timepoint. Moreover, this study focused on characterizing the potential multiple roles of JA during aphid infestation in altering plant physiological traits to benefit the aphids itself. Therefore, we chosen time courses at the endpoint (9 or 24 h). In this time courses, we could detect the potential multifunction of JA accumulation during aphid infestation on altering the plant physiological traits that suppress the plant defenses in wheat cultivar XN979. Please allow us still using the time courses at the endpoint (9 or 24 h).

For Figures 6 and 7: Explain what "0h" represents. To my understanding, "0h" means without aphid treatment; however, all genes were significantly upregulated in the pre-infested group compared to the control. Please clarify.

Response: In the present study, "0h" meant that all aphids removal from seedling at that timepoint after 24 h of aphid infestation. We guess the sentence “leaf tissue in aphid infestation treatments were collected at 0 h, 6 h, 12 h, and 24 h post-aphid removal” could express our meaning.

In addition, it is unclear why the MeJA/SA treatments were separated from aphid infestation treatment in those gene expression measurements, and samples were collected at different time points. It would be better to have a side-by-side comparison of the gene expression between MeJA/SA treatments and aphid infestation.

Response: As mentioned previously, this study focused on characterizing the potential multiple roles of JA during aphid infestation in altering plant physiological traits to benefit the aphids itself. Thus, in the first part, we directly treated the wheat plants with MeJA to simulate the JA condition in plant tissues, then it would alter plant physiological traits. After that, we measured the transcriptional profiles of marker genes after aphids infestation. Both the treatments exhibited the similar profiles of marker genes, that means the plant physiological alternations might be triggered by JA during aphid infestation. Moreover, in comparison with direct MeJA treatment, aphid infestation treatment had a 24 h infestation, during the feeding, the plants would trigger the phytohormones biosynthesis. So we immediately collected the plant tissue after aphid removal at 0 h, while in the direct MeJA treatment, the plant need a short time to absorb the MeJA, and MeJA need a short time to trigger the plant physiological alternation, so we collected the plant tissue since 3 h post MeJA treatment. Therefore, we separately analyzed the gene expression profiles between MeJA treatments and aphid infestation treatment at different sampling timepoints.

For all experiments, overall discussion, and conclusion:

The authors have discussed how aphid infestation changes the gene expression in JA/SA signaling pathways and how those pathways could influence aphid performance, and potentially how the intermediates (auxins or serotonin) could be involved in the interactions between the host plant and aphids. However, none of those plant hormones were measured in any of the experiments. Measuring those hormones would bridge the gap between the current results and the authors' intended conclusions.

Response: Yes, it better for measuring the level of those hormones in the plant tissue that would bridge the gap of uncertain results. Thank you very much for your kind suggestion. However, those hormonal compounds would be quickly participant in different activities in plant tissue after it formed. For instance, MeJA could be quickly conjugated with isoleucine (ILE) to form JA-ILE and active the JA signaling. Thus, it is hard for accurately measuring the level of phytohormones in plant tissues. In future, we’ll conduct more experiment to accurately measure the level of phytohormones in plant tissues.

Figure 8 needs a higher resolution image, and the intermediates (chemicals such as serotonin, as proposed by the authors) need to be added to the model.

Response: We apologize for not uploading the higher resolution figure 8 in the original submission. To improve the readability and logic of the figures, in this manuscript, we modified the text and the arrows in the figures and added more content the descriptions in the figure legends.

I hope my comments could be helpful in significantly improving the manuscript.

Response: Thank you very much for your kind suggestion on improving the manuscript. We carefully revised the manuscript following your suggestion and comments.

Reviewer 2 Report

The manuscript on “Previous Infestation with Sitobion miscanthi Induces Jasmonic Acid Accumulation that Suppress Plant Defense Responses in Wheat Cultivar XN979 is an interesting work on wheat defenses.

You need to explain why you conduct this research, how it is going to be applicable in the wheat agroecosystem. I see only the antagonistic interaction between two signaling pathways. Is there any ecological relevance to this study? You didn’t introduce the functions/ importance of the selected genes in SA/JA pathways. Please include a paragraph on this. So the reader can easily understand the manuscript. The results supported a further understanding of the antagonistic interaction between hormone signaling in cereals under aphid infestation. The author conducted various experiments to support this. The manuscript was disorganized, especially in the introduction, methods, and results. Some of the methods were not sufficiently described. Some statistical analysis is not appropriate and doesn’t describe it well. The overall discussion is written well. I see a lot of time font changes throughout the text. Please use one font style and size.

In the following, I will point out some (but due to time constrain not all) of specific points:

Simple summary:

I am not sure this is needed for the manuscript.

Abstract:

Line 26: Should be wheat cultivar.

Line 29: Briefly explain what parameters are affected by MeJA.

Line 33: What is TaTDC 33 stand for? Which pathway is affected by this gene.

Introduction:

Line 43-44: delete favorite food crop. It depends on the food habits of the region.

You mention about variety of wheat pests. Here is an example for another significant pest in wheat agroecosystem:

Weeraddana et al., 2021. A laboratory method for mass rearing the orange wheat blossom midge, Sitodiplosis mosellana (Diptera: Cecidomyiidae)

Line 49: provide some examples of viral diseases spread by s Sitobion avenae.

Line 64: SA-JA pathways are mostly antagonistic; however, there are some instances recorded as synergistic. Provide an example to this statement.

You need to include a paragraph on the ecological relevance of this study . . .

Also, you didn’t mention the MeSA and MeJA, which are the key mobile signals of plant communications. Explain and provide examples.

Line 85-86: Cite any relevant studies which used molecular targets for breeding wheat cultivars.

Methods:

Line 98: how old the wheat seedlings are? Need more information

Line 106: What type of soil is mentioned here?

Line 112: cite a reference for the wheat growth stages described in the manuscript.

Line 119-120: Why did you choose these SA or JA concentrations? Are there any references to cite? How do you avoid inducing plant-plant communication following SA/JA treatments?

Line 132-143: check the font sizes and font type.

Line 140-143: Not clear, rewrite the sentences.

Line 140-155: Provide how you calculate these population parameters or cite references explaining these equations.

Line 158-159: Did you collect the foliar tissues locally or systematically for the aphid infestation and hormone treatment?  Please clearly state it.

You need to include what genes you look for in this study.

Line 186-187: Using three technical replicates to calculate the mean and error is incorrect. Maybe use the average.

Statistical analysis:

Did you check the normality and variance before conducting the parametric tests?

I wouldn’t use statistical analysis on gene expression. Why did you choose Newman over the Tuckey test?

Results:

How do you get the count of eggs? Include more description on this in the method. Can you include different line types so that everyone can see the difference? Did you check the overall difference throughout all time points? I think there might be some based on Fig1. Overlapping error bars makes the graph unclear. Please make some necessary changes to Fig1.

 Table 1:

What do you mean by CK here? You don’t need both a table and five figures to provide results. I would use only the figures.

Fig3: I would remove the error bars as you use technical replicates. I will not use statistical analysis on technical replicates.

Remove over control from Y axis of Fig 4,5,6 and 7.

Discussion:

Overall, your discussion is written well. I have a few minor additions, as mentioned below.

Figure 8: That’s a good figure but font is too small to read.

Line 411-412: See another example from a different agroecosystem (canola) here. Infection with clubroot pathogen, a biotrophic pathogen-induced SA pathway, suppressed the bertha armyworm's oviposition in canola.

Weeraddana et al 2021. Infection of canola by the root pathogen Plasmodiophora brassicae increases resistance to aboveground herbivory by bertha armyworm, Mamestra configurata Walker (Lepidoptera: Noctuidae).

The overall quality of the English writing is good. 

Author Response

Thank you very much for your letter and the comments about our manuscript submitted to your journal. We would also like to thank the reviewers for their careful and constructive reviews. Soon after receiving your comments, the other authors and I carefully revised the manuscript. The changes made according to the comments from the reviewers are detailed below.

The manuscript on “Previous Infestation with Sitobion miscanthi Induces Jasmonic Acid Accumulation that Suppress Plant Defense Responses in Wheat Cultivar XN979 is an interesting work on wheat defenses.

Response: Thank you very much for carefully reviewing our manuscript and kindly giving us many suggestions and comments for improving the readability of this manuscript. In the revision, we carefully followed your corrections.

You need to explain why you conduct this research, how it is going to be applicable in the wheat agroecosystem. I see only the antagonistic interaction between two signaling pathways. Is there any ecological relevance to this study? You didn’t introduce the functions/ importance of the selected genes in SA/JA pathways. Please include a paragraph on this. So the reader can easily understand the manuscript. The results supported a further understanding of the antagonistic interaction between hormone signaling in cereals under aphid infestation. The author conducted various experiments to support this. The manuscript was disorganized, especially in the introduction, methods, and results. Some of the methods were not sufficiently described. Some statistical analysis is not appropriate and doesn’t describe it well. The overall discussion is written well. I see a lot of time font changes throughout the text. Please use one font style and size.

Response: Thank you very much for your comments and advices. Because this study focused on characterizing the antagonistic interaction between JA and SA signaling pathways in wheat plants during preinfestation with cereal aphids. Thus, the introduction and discussion section of this study focused more on illustrating the biological roles of marker genes involved in the JA- and SA-signaling. Although there were a lot of experimental evidence that referred to the defense responses to against the pests, the similar defense responses could be detected, such as the system acquired resistance, hypersensitive responses that were triggered in SA signaling. So we didn’t pay much attention on illustrating the exact plant responses after diverse pests’ damage. In addition, in recently years, we conducted a couple of studies that focused on the phytohormones including the defenses hormones (SA and JA), and the growth hormones (Auxins and serotonin) mediated the interactions between crop plants and different species of wheat invaders. For instance, after fungi invasion, the plant would extensively accumulate a large amount of IAA in wheat cultivar head tissues 4 d after initial infection, however, in the feeding experiment, we could not detect any auxins when provide the fungi with L-TRP. It was demonstrated that infection by fungi usually leads to increased biosynthesis of L-TRP in wheat and Brachypodium distachyon. The experimental evidence in wheat-fungi and rice-planthopper systems suggested that the extensive accumulation of JA in plant tissue after attacks could lead to the production of auxins and serotonin. Accordingly, we supposed that the environmental alterations in plant tissues triggered by plant immune systems synergistically facilitate the extensive accumulation of auxin from L-TRP in host plant tissues, and we didn’t illustrate the experimental evidence in other plant-pest systems. Moreover, because the potential role of plant growth hormones (including auxins and serotonin) have multiple pathways to trigger the environmental alterations in host plants, and some of them involved in the fine-tune the SA dependent responses. Thus, we focused more on illustrating the JA signaling pathways and its derived compounds mediated pathways to suppress the SA-mediated defense responses. Based on this, we had established the potential pathways of plant growth hormones accumulation in wheat, and it would provide the molecular targets for wheat cultivars improvements. So, please forgive us not follow some of your comments and advices on the introduction and discussion section.

In addition, we selected some of important marker genes associated with the JA- and SA-signaling, which concerned by most of the studies. Thus, in the original submission we didn’t illustrate its function either in methods or results section. For facilitating the readers to understanding the manuscript, in the revision, we classified the marker genes concerned in this study by their main function in the method sections. Moreover, we’re very sorry for we didn’t rationally construct the structure of the manuscript and stringently express some of the methods in details in the original submission. To improve the readability and logic of the manuscript, after careful consideration, we rephrased or revised some parts strictly following your suggestion and comments, especially in the methods sections we provided more details about what we conducted. Please allow us to make the corrections and modifications.

In the following, I will point out some (but due to time constrain not all) of specific points:

Response: Yes, in the revision, we carefully followed your corrections.

Simple summary:

I am not sure this is needed for the manuscript.

Response: Yes, it should be provided. When I browsed the original research article, I found most of them illustrated the Simple summary section prior Abstract.

Abstract:

Line 26: Should be wheat cultivar.

Response: Thank you very much for the rephrase suggestion. We have placed “wheat cultivar” at there.

Line 29: Briefly explain what parameters are affected by MeJA.

Response: For the limitation of the length of Abstract, we didn’t provide the exact name of parameters that MeJA affected, because each of them have a long name. In the revision, we rephrased that sentence as “The preference experiment and the life-table data revealed that direct foliage spraying of 2.5mM Methyl Jasmonate (MeJA) exhibited the weak negative or positive effects on the preference selection, and the population dynamics and oviposition parameters of S. miscanthi.” Please allow us to make the modifications.

Line 33: What is TaTDC 33 stand for? Which pathway is affected by this gene.

Response: We provided the full name of TaTDC gene and its associated with exact pathway in plant tissues. In the revision, we rephrased that sentence as “the transcript level of tryptophan decarboxylase (TaTDC) gene had rapidly upregulated after the treatments as well, which facilitate the conversion of L-tryptophan to tryptamine.”

Introduction:

Line 43-44: delete favorite food crop. It depends on the food habits of the region.

Response: We deleted the phase “favorite food crop” from that sentence.

You mention about variety of wheat pests. Here is an example for another significant pest in wheat agroecosystem:

Weeraddana et al., 2021. A laboratory method for mass rearing the orange wheat blossom midge, Sitodiplosis mosellana (Diptera: Cecidomyiidae)

Response: It’s true that the orange wheat blossom midge is another species of wheat pests. In comparisons with wheat aphids, this pest only attacks the wheat blossom, while the wheat aphids could damage the wheat plants during the entire growth period. In addition, this study focused more on characterizing the physiological alternation in wheat leaf tissues induced after JA spraying or aphid infestation. Thus, we didn’t illustrate this species of pest in the introduction. Please forgive us not following your suggestions.

Line 49: provide some examples of viral diseases spread by s Sitobion avenae.

Response: We provided the virus that S. avenae could transmit in agroecosystem. In the revision, we rephrased that sentence as “The wheat aphid, Sitobion miscanthi Takahashi (Hemiptera: Aphididae), is one of the most destructive and common piercing-sucking herbivores insects that seriously attack the cereal plants by directly sucking the photoassimilates, and transmitting the crop viruses, such as barley yellow dwarf virus in the field agroecosystem.”

Line 64: SA-JA pathways are mostly antagonistic; however, there are some instances recorded as synergistic. Provide an example to this statement.

Response: It’s true that there were experiment evidence revealed that SA-JA pathways act as synergistic in some plant-pest systems. This study focused more on characterizing the physiological alternation in wheat leaf tissues induced after JA spraying or aphid infestation, which probably trigger the antagonistic effects on SA-mediated defense responses or compenetrate the adverse effects of host defense responses on the fitness of aphids. Thus, we didn’t illustrate the synergistic instances SA-JA pathways in the introduction. Please forgive us not following your suggestions.

You need to include a paragraph on the ecological relevance of this study . . .

Response: Although we didn’t mention the ecological relevance of this study in the introduction section, we talked about the ecological effects of the crosstalk of SA-JA pathways in the discussion section.

Also, you didn’t mention the MeSA and MeJA, which are the key mobile signals of plant communications. Explain and provide examples.

Response: It’s true we didn’t mention the MeSA, because SA is the common compound that could be get in a cheap price, thus, most of studies using SA to direct exposure to the plants, so did in the current study. In comparisons with SA, most of studies using MeJA to stimulate the effects of JA on the plants, because the JA compound is more expensive than MeJA. Thus, either in introduction or methods section, we didn’t mention the MeSA.

Although there were a diverse of studies focused on characterizing the plant defenses that triggered by phytohormone accumulation during pests attacks, as far as I am concerned from available literatures that the plant defenses didn’t

Line 85-86: Cite any relevant studies which used molecular targets for breeding wheat cultivars.

Response: Those relevant studies had been illustrated in the discussion section.

Methods:

Line 98: how old the wheat seedlings are? Need more information

Response: Yes, we provided more detail of the age of wheat seedlings in the revision.

Line 106: What type of soil is mentioned here?

Response: Yes, we provided more detail of the type of the soil in the revision.

Line 112: cite a reference for the wheat growth stages described in the manuscript.

Response: In this study, we employed the BBCH-scale to describe the the wheat plant developmental stage. Yes, we provided the reference about the BBCH-scale in the methods section.

Line 119-120: Why did you choose these SA or JA concentrations? Are there any references to cite? How do you avoid inducing plant-plant communication following SA/JA treatments?

Response: Prior the experiment, we conducted a pre-experiment to explore the suitable concentrations in the following experiment. In the pre-experiment, we set a range of JA concentrations from 0.25, 0.5, 1.0, 1.5, 2.0, 2.5, 3.0, 4.0 mM. According to the results of the pre-experiment, we found that direct foliage spraying of 2.5 mM could significantly upregulated the transcript level of TaTDC gene, which facilitate the conversion of L-tryptophan to tryptamine. Thus, we selected this concentration of JA or SA to spraying the wheat foliage. In addition, when we spraying the phytohormones, we separated them in different room, after drying for 6 h, the wheat plants with different phytohormone treatments were put into the same growth chamber. We guess we could avoid the plant-plant communication by reducing the volatilization in this way.    

Line 132-143: check the font sizes and font type.

Response: Sorry for that, we had corrected the errors about the font sizes and font type.

Line 140-143: Not clear, rewrite the sentences.

Response: Sorry for that, we had rewritten that sentence.

Line 140-155: Provide how you calculate these population parameters or cite references explaining these equations.

Response: We apologize for not precisely described the definitions, explanations, and equations of the parameters used in the life table analysis. Because there were a diverse of studies that referred those information, thus we didn’t illustrate them in the original submission. In the revision, as mentioned previously, for omitting the extra information, we cited a reference that illustrated these information in detail. Please allow us to make such modification in the revision.

Line 158-159: Did you collect the foliar tissues locally or systematically for the aphid infestation and hormone treatment?  Please clearly state it.

Response: Because the wheat plant tissues were collected at the three-leaf stage (BBCH 13), thus it is hard to determine the effects of phytohormones triggered is local or systematical. Therefore, in this study, we just collected the foliar tissue from the third leaf in each seedlings.

You need to include what genes you look for in this study.

Response: Sorry for that, as mentioned previously, we selected some of important marker genes associated with the JA- and SA-signaling, which concerned by most of the studies. Thus, for facilitating the readers to understanding the manuscript, in the revision, we classified the marker genes concerned in this study by their main function in the method sections.

Line 186-187: Using three technical replicates to calculate the mean and error is incorrect. Maybe use the average.

Response: Sorry for that mistake, it should be biological replicates.

Statistical analysis:

Did you check the normality and variance before conducting the parametric tests?

Response: Yes, prior to conduct the one-way ANOVA, we had conducted the homogeneity test for each parameters.

I wouldn’t use statistical analysis on gene expression. Why did you choose Newman over the Tuckey test?

Response: In the analysis of gene expression profiles, we were used the Student's t test to determine the difference of transcript level of marker genes between the JA treatment or aphid infestation and healthy plants. For comparing the colonization number and oviposition of aphids between phytohormones and control, we had conducted the homogeneity test for each parameters, then conducted the one-way ANOVA to compare the difference between treatments. If there was significant difference between different treatments, we conducted the multi-comparisions of mean value of different parameter with the SNK. Why we chosen Newman over the Tukey test? Because SNK methods is a more powerful method than Tukey, and it is more suitable for the exploratory studies. In addition, in our previous studies, we used to adopting this method to compare the mean value of different parameters associated with the performance of aphids.

Results:

How do you get the count of eggs? Include more description on this in the method. Can you include different line types so that everyone can see the difference? Did you check the overall difference throughout all time points? I think there might be some based on Fig1. Overlapping error bars makes the graph unclear. Please make some necessary changes to Fig1.

Response: We daily counted and recorded the number of aphid nymphs on each seedlings, after that, we removed all of them individually. The next day, we counted and recorded it again until the sixth day post adult aphid infestation. Because we paied more attention on characterizing the physiological alternations triggered by JA sparying, thus we didn’t conduct the preference experiment on other wheat cultivars. In addition, mmong of these sampling timepoints, most of them exhibited the similarly profiles on aphid colonization, thus we didn’t conducted the overall difference throughout all time points on the parameters in preference experiment. Sorry for that. Moreover, we prepared the new figure 1 followed your comments.

 Table 1:What do you mean by CK here? You don’t need both a table and five figures to provide results. I would use only the figures.

Response: CK mean the control. We had used control instead of CK in new Figures and table.

Fig3: I would remove the error bars as you use technical replicates. I will not use statistical analysis on technical replicates.

Response: As mentioned previously, sorry for that mistake, it should be biological replicates.

Remove over control from Y axis of Fig 4,5,6 and 7.

Response: Because the relative expression level of marker genes were calculated with the 2–ΔΔCt method, for calculating the transcript level of genes, the Ct value of the control had been considered in that. Thus we guess the name of Y axis of Fig 4,5,6 and 7, and we didn’t follow your suggestion. Please allow us to keep it in those figures.

Discussion:

Overall, your discussion is written well. I have a few minor additions, as mentioned below.

Figure 8: That’s a good figure but font is too small to read.

Response: We apologize for not uploading the higher resolution figure 8 in the original submission. To improve the readability and logic of the figures, in this manuscript, we modified the text and the arrows in the figures and added more content the descriptions in the figure legends.

Line 411-412: See another example from a different agroecosystem (canola) here. Infection with clubroot pathogen, a biotrophic pathogen-induced SA pathway, suppressed the bertha armyworm's oviposition in canola.

Weeraddana et al 2021. Infection of canola by the root pathogen Plasmodiophora brassicae increases resistance to aboveground herbivory by bertha armyworm, Mamestra configurata Walker (Lepidoptera: Noctuidae).

Response: Thank you very much for your kind suggestion of adding the contents experiment evidence from canola (Brassica napus L.) and its parasites. These results provided the experimental evidence that SA accumulating after pathogen infection probably suppress the JA accumulation and its positive role in plant growth. Thus, we followed your suggestion and added this contents in the discussion section.

Round 2

Reviewer 2 Report

Dear Author,

Thank you for responding to my queries.  It is easy for me to follow your responses if you mention the line numbers to which you provided the answers. I am satisfied with the changes made in the manuscript.

Author Response

We would like to thank you again for your careful and constructive reviews. You know that the Editorial Office of the Journal asked us to revise the manuscript in track mode, there are two mode of the tracking file, thus it is hard for us to determine which one could be illustrated for reviewer's file. Therefore, we didn't illustrate the line number in the response file. Sorry for that.